# Reminiscent capillarity in subnanopores

Irena Deroche [1,2]*, T. Jean Daou [1,2], Cyril Picard[3] & Benoit Coasne [3]*

Fluids in large and small pores display different behaviors with a crossover described through the concept of critical capillarity. Here we report experimental and simulation data for various siliceous zeolites and adsorbates that show unexpected reminiscent capillarity for such nanoporous materials. For pore sizes $D$ exceeding the fluid molecule size, the filling pressures $p$ are found to follow a generic behavior $k_B T \ln p \sim \gamma/\rho D$ where $\gamma$ and $\rho$ are the fluid surface tension and density. This result is rationalized by showing that the filling chemical potential for such ultra-small pores is the sum of an adsorption energy and a capillary energy that remains meaningful even for severe confinements. A phenomenological model, based on Derjaguin's formalism to bridge macroscopic and molecular theories for condensation in porous materials, is developed to account for the behavior of fluids confined down to the molecular scale from simple parameters.

[1] Université de Haute Alsace, CNRS, ISMM, UMR 7361, 68057 Mulhouse, France. [2] Université de Strasbourg, 67081 Strasbourg, France. [3] Université Grenoble Alpes, CNRS, LIPhy, 38000 Grenoble, France. *email: irena.deroche@uha.fr; benoit.coasne@univ-grenoble.-alpes.fr

Confinement of fluids in porous media involves a diversity of phenomena such as physical adsorption, chemical reactions, and solubilisation which give rise to complex behaviors including wetting/adhesion, nucleation, slippage, surface diffusion, surface reconstruction, etc. Owing to their ultra-small pore size $D$ and large surface area, nanoporous ($D \sim$ nm) and subnanoporous ($D <$ nm) solids such as zeolites, active carbons, and metal organic frameworks constitute a critical subfamily of the broad class of porous materials with specific applications in phase separation, catalysis, etc.[1–3]. Yet, despite their increasing role in fundamental and applied science, the behavior of fluids in these materials still remains only partially explored by many aspects. Predicting the thermodynamical equilibrium of a given fluid in one of these nano/subnanoporous hosts requires to carry out on purpose specific experiments or molecular simulations as there is no general macroscopic theory relying on a simple set of known experimental parameters (e.g., density, surface tension, etc.). From a fundamental standpoint, confinement in porous solids with subnanometric to nanometric cavities was first described by extending classical adsorption and capillarity theories[4,5]. In practice, while physical models such as the celebrated Langmuir, BET or Kelvin equations were found to be qualitatively valid for such small pores, it was soon realized that their use in this specific context remains essentially empirical (for instance, the concept of independent adsorption sites and adsorbed layers in such ultraconfining media is clearly inconsistent with their geometry and atomic structure).

The advent of molecular theories such as the Density Functional Theory (DFT) in statistical mechanics[6,7] and atom-scale simulations has allowed describing the physics of fluids in nanoporous media and establishing a bridge with the classical thermodynamics for large pores[8–10]. From this new era, a unified picture has emerged with a crossover between irreversible capillary condensation—i.e., first order transition—for large pores and reversible, continuous pore filling—i.e., second order transition—for small pores (Fig. 1)[6,10–13]. On the one hand, for large pores $D \gtrsim 10\sigma$ ($D$ and $\sigma$ are, respectively, the sizes of the pore and of the confined molecules), pore filling first involves the formation of an adsorbed film followed by hysteretic capillary condensation at a pressure smaller than the bulk saturating vapor pressure $p_0$. On the other hand, for small pores $D \sim \sigma$, pore filling does not involve a well-defined confined gas/liquid interface and occurs through a continuous and progressive density increase of the confined fluid. The crossover between these two asymptotic limits is described through the concept of the capillary critical temperature $T_{cc}$ which is shifted with respect to the bulk critical temperature $T_c$ (Fig. 1); for a given pore size $D$, capillary condensation occurs for $T < T_{cc}(D)$ while pore filling is reversible and continuous for $T > T_{cc}(D)$. Reciprocally, for a given $T$, capillary condensation occurs for pore sizes $D > D_c$ while filling is reversible and continuous for $D < D_c$ where $D_c \sim 4\sigma T_c/(T_c - T_{cc})$[10,12]. Despite the comprehensive picture above, a macroscopic theory capable to predict the quantitative thermodynamic behavior (typically the pore filling pressure) of fluids in ultraconfining media is still missing. Such a theory is highly desirable as available molecular tools suffer from the following limitations. On the one hand, atom-scale simulations are efficient but time-consuming and, more importantly, forcefield-dependent so that large departure with respect to experiments cannot be ruled out. On the other hand, Density Functional Theory is more practical but, being a mean-field theory, it requires to calibrate interaction parameters against some already existing available experiments.

Here, we use experimental and molecular simulation data to establish a simple macroscopic model that allows predicting the behavior of any confined fluid in subnanoporous and nanoporous solids using simple parameters such as density and surface

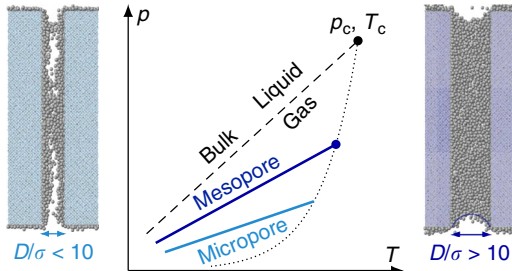

**Fig. 1** Criticality in confined fluids. Shift in the gas/liquid transition for a fluid having a molecular size $\sigma$ confined in a nanopore of a diameter $D$. On the one hand, in mesopores, typically for $D \gtrsim 10\sigma$, the fluid can be treated using a continuum model which rests on capillarity ingredients such as the surface tension, radius of curvature of the gas/liquid interface, etc. (*right*). On the other hand, in micropores, typically for $D \lesssim 10\sigma$, the molecular granularity of the fluid has to be taken into account and continuum models no longer apply (*left*). For a given pore size $D$, the bulk gas/liquid coexistence (dashed line) and the bulk critical point $p_c$, $T_c$ (black sphere) are shifted to lower pressure and temperature with a scaling that depends on $D$. In particular, upon confinement in a pore of a size $D$, the critical point $T_c$ is shifted to a lower value $T_{cc}$ such that $\Delta T_c = T_c - T_{cc} \sim T_c \sigma/D$

tension. Non-polar and polar fluids with drastically different bulk properties are used to obtain a fluid-independent, general picture when confined in such solids. Siliceous zeolites (the so-called zeosils) are considered as they exhibit pores of a regular geometry combined with a simple surface chemistry and therefore allow developing a macroscopic phenomenological model. We first show that, unless pores are smaller than the confined fluid size $D < \sigma$, all our data follow a master behavior in which the chemical potential is proportional to a capillary energy, $\mu \sim \gamma/\rho D$, but is shifted by an offset corresponding to an adsorption energy. This important finding, which suggests a reminiscent capillary behavior even in ultra-confining pores, allows bridging the gap between molecular theories relevant to small pores and capillary concepts relevant to macroscopic pores. From this observation, we build on Derjaguin's theory for adsorption and capillary condensation[14–17] to extend its framework to subnanoporous media. We show that its unexpected applicability to such materials arises because the following asymptotic limit is reached. Upon decreasing $D$, the filling pressure tends to $p \sim 0$ with a corresponding chemical potential $\mu \sim k_B T \ln p$ that is given by the sum of a capillary contribution and an adsorption contribution. The first contribution $\mu_{cap} \sim \gamma/\rho D$ corresponds to the chemical potential predicted from the surface tension and the surface to volume ratio using the capillarity theory taken in the limit of vanishing adsorption (i.e., the film thickness is vanishingly small). The second contribution $\mu_{ads} \sim f(t, \xi)$ is an adsorption contribution, dependent on the surface interaction range $\xi$ and film thickness $t$. As expected, for nanoporous and subnanoporous media, this adsorption contribution is by no means negligible compared to the capillary contribution. Yet, its variations when considering the different fluids, temperatures, etc. used here remain small compared to the capillary contribution so that the latter mostly governs the chemical potential at pore filling. A large body of experimental data taken from the literature confirm the simple picture underlying the proposed model. This phenomenological approach presents some limitations (for instance, when solids with significant surface heterogeneity are considered as it leads to far more scattered adsorption contributions in the chemical potential at pore filling) but its simplicity and robustness makes it a very promising tool in fields where ultra-confining media are relevant (phase separation, catalysis, membrane science in chemistry/chemical engineering, nanofluidics and energy

storage/conversion in physics, depollution and fluid transfer in earth, and soil science, etc.).

## Results

**Adsorption for various zeosils and adsorbates.** Different molecules were considered to probe a broad range of zeolite/fluid interactions (Fig. 2): acetone (a highly polar oxygenated hydrocarbon), *n*-hexane (a weakly polar aliphatic hydrocarbon), *p*-xylene (a non-polar aromatic hydrocarbon with aliphatic substitution) at room temperature and nitrogen at 77 K (a simple molecular probe used for routine characterization of porous materials). Details of the experimental measurements can be found in the Methods section. Some important physico-chemical properties of these molecular probes are summarized in Supplementary Table 1. Similarly, several representative zeolites were considered in this work (Fig. 2): beta zeolite (*BEA)[18], silicalite-1 (MFI)[19], chabazite (CHA)[20], and STT[21]. The three-dimensional channel networks in *BEA, MFI, and CHA are delimited by 12, 10, and 8 membered-rings, respectively, while the two-dimensional channel network in STT is made up of odd openings formed by 7 and 9 membered-rings. The main structural parameters of these zeolites are reported in Supplementary Table 2 while an additional view of their porous network structure is shown in Supplementary Fig. 1.

Gas adsorption in these siliceous zeolites was assessed for the four adsorbates using Monte Carlo simulations in the Grand Canonical Monte Carlo ensemble (GCMC)[22]. Like in a real experimental adsorption set-up, this statistical mechanics technique allows considering a host porous medium having a constant volume $V$ in equilibrium with a bulk reservoir of gas molecules that imposes its temperature $T$ and chemical potential $\mu$. Once equilibrium is reached, the adsorbed amount $n(\mu, T)$ at a given $\mu$ and $T$ is readily obtained as an ensemble average of the number of molecules in the zeolite. The adsorption isotherm $n(p, T)$ is then plotted by converting $\mu$ into gas pressure $p$. For each gas, the relationship $\mu(p)$ was determined using the Widom insertion method in the course of $NpT$ Monte Carlo simulations. Such simulations are efficient for bulk phases as considered here (since the relationship $\mu(p)$ has to be determined for the bulk fluids only). As for the adsorption simulations using the GCMC algorithm, especially for complex fluids such as some of the fluids considered here, it is known that proper equilibration to reach the physical density of the confined fluid suffers from technical limitations. More in detail, due to the intrinsic difficulty in inserting molecules into the small pores of nanoporous solids, GCMC can be very slowly converging to reach the equilibrium density. To circumvent poor sampling efficiency and guarantee convergence towards equilibrium, our GCMC molecular simulations were performed using the Configurational-Bias algorithm[23]. All details regarding the applied computational technique and the molecular models used for the different gases and zeolites are described in the Methods section. A detailed analysis of the simulated data can be found in Supplementary Figs. 3–5 and Supplementary Tables 4 and 5 together with a thorough comparison with available experimental and simulation data in the Supplementary Discussion (including a comparison of adsorbed amounts, Henry constants, isosteric heats of adsorption). The supplementary information also contains the structure file for each of the 4 zeolite structures (in .car format).

The adsorption isotherms for the different gases and zeolite structures are shown in Fig. 3. As expected for subnanoporous materials such as zeolites, all adsorption isotherms are of type I—Langmuir-like—according to IUPAC classification; the adsorbed amount increases rapidly at very low pressure in a continuous and reversible fashion (note the use in Fig. 3 of a log scale for the

pressure axis to better display the very low pressure range). As shown in Supplementary Fig. 6, in agreement with the experimental data, for each fluid and zeolite, the plateau reached at saturation can be correctly predicted from the known porous volume of the zeolite if the density of the confined adsorbate is taken equal to the bulk density. This relation, known as Gurvich's rule[5], suggests that the molecular simulations have reached equilibrium and that the configuration-bias employed in the framework of GCMC simulations allows efficient phase space sampling. From a practical viewpoint, such a simple scaling provides a simple and rapid means to estimate adsorption capacities for any adsorbate/adsorbent couple.

**Generic scaling and master curve.** Going back to the adsorption isotherms in Fig. 3, rapid inspection of the data suggests that there is no systematic pore size dependence as the order in pore filling pressures for the four zeolites is not identical for the different adsorbates. For each system, the filling pressure $p_f/p_0$ was estimated as the pressure at which pores get half-filled. This definition is a robust way to characterize the position of the sharp variation of the pore content revealed in semi-log plot. While other definitions could be used, they would not change the outcome of our discussion below. In spite of the apparent complexity in linking the pore size and the pore filling pressure for the different zeolite/adsorbate couples, the thermodynamics ruling adsorption in such subnanoporous media can be rationalized as follows. As shown in Fig. 4, for pores and adsorbates such that $D/\sigma > 1.4$, i.e., for which the adsorbate molecule fits inside the host porosity, a simple scaling emerges between the chemical potential at which pore filling occurs, $\mu - \mu_0$ and the capillary energy $\gamma/(\rho D)$. More precisely, $\mu - \mu_0$ is the shift in chemical potential with respect to its value $\mu_0$ at the bulk saturating vapor pressure $p_0$. Each adsorbate was assumed to behave as an ideal gas, i.e., $\mu \sim RT \ln p$, which is a reasonable approximation since pore filling occurs at extremely low vapor pressures. The capillary energy, $\gamma/(\rho D)$, which is directly proportional to the Laplace pressure, i.e., the difference between the pressures in the bulk gas phase and in the nanoconfined liquid $\Delta p \sim \gamma/D$, describes the driving force responsible for capillarity in pores. The molecular simulation data in Fig. 4 follow the same scaling with a simple linear relationship between chemical potential and capillary energy (note that pores such that $D/\sigma < 1.4$ deviate from this simple scaling because of strong repulsion/friction with the host zeolite for such bulky adsorbates). Interestingly, experimental data from the literature—which were chosen for other zeolites and fluids—nicely fall onto the same master curve, therefore validating the unique relationship observed in the present work. In particular, the data used to establish such a correspondence between chemical potential at pore filling and capillary energy correspond to different fluid molecule shapes—small regular versus chain molecules—with chemical structures being either polar or apolar. Moreover, both the simulated and experimental data used to build the plot in Fig. 4 were obtained for fluids at different temperatures (typically, at room temperature and/or at temperature of boiling nitrogen). The fact that all these data follow the same trend described by the simple scaling above further supports the underlying theoretical picture. As a further validation, the data shown in Fig. 4 also include a data point obtained for a non-silica zeotype—Ar at 77 K in the alumino-phosphate $AlPO_4$-5—which suggests that this scaling also applies to other members of this broad porous solid family. The use of the chemical potential shift $\mu - \mu_0$ instead of the filling pressure $p_f/p_0$ is justified by the fact that the former is the appropriate driving force for capillary condensation/density change (since the chemical potential is the conjugated variable of the number of

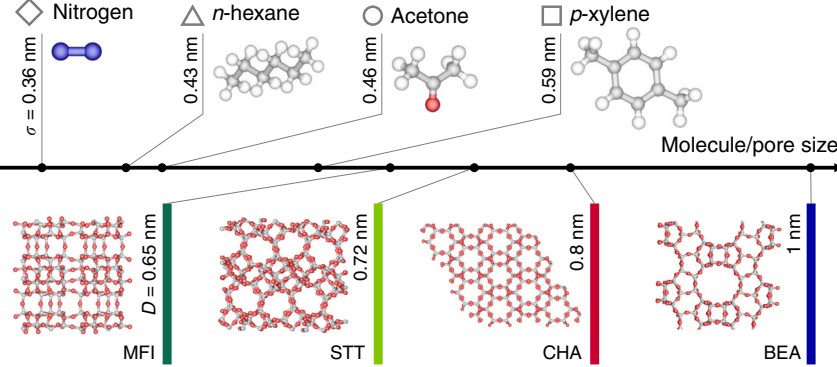

**Fig. 2** Fluids and nanoporous materials. Different fluids and zeolite matrices considered in this paper sorted according to their size ($\sigma$ for the fluid molecule and $D$ for the zeolite pore). Throughout the paper, solvents (upper part) are denoted according to the following symbol code: $N_2$ (lozenges), $n$-hexane (triangles), acetone (circles), and $p$-xylene (squares). For each fluid, the molecular size $\sigma$ was taken as the kinetic diameter which is estimated to match the second virial coefficient. This choice is particularly suitable as it applies both to fluids in gas-like and liquid-like states. The zeolite matrices (lower part) are denoted according to the following color code: silicalite-1 (MFI, dark green), STT (light green), chabazite (CHA, red), and beta (BEA blue). For each zeolite, the pore size was estimated from the fractional free volume as determined using the Connolly surface area method (Supplementary Fig. 2)

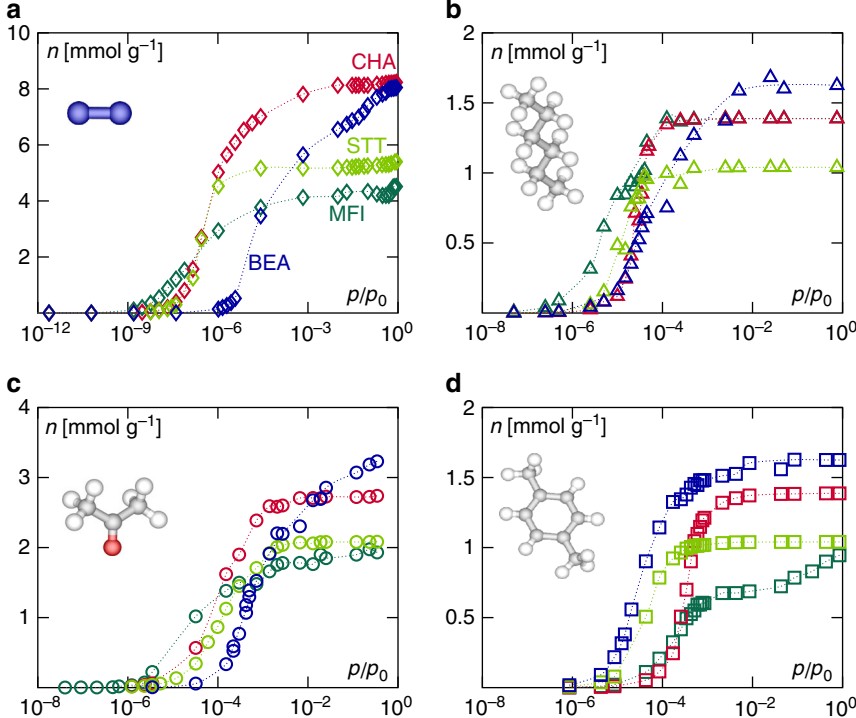

**Fig. 3** Fluid adsorption in siliceous zeolites. Simulated adsorption isotherms for nitrogen at 77 K (**a**), $n$-hexane at 298 K (**b**), acetone at 298 K (**c**), and $p$-xylene at 298 K (**d**) in different siliceous zeolites: silicalite-1 (MFI, ortho, dark green data), chabazite (CHA, red data), STT (light green data), and beta (*BEA, blue data). Note the use of a log scale for the abscissas which correspond to the gas pressure axis. The bulk saturating vapor pressure $p_0$ for each gas at the corresponding temperature is given in Supplementary Table 1

molecules in the grand free energy). However, in the meantime, the use of a log scale $\mu \sim RT \ln(p/p_0)$ introduces a non-negligible uncertainty when using the master curve shown in Fig. 4 to predict filling pressures. At worse, the relative uncertainty over the chemical potential at pore filling is about 25% which leads to the same relative uncertainty for the log of the filling pressure. While this seems reasonable given the broad applicability of the master curve provided here, this uncertainty should be included when rigorously estimating the filling pressure for a specific example.

**Reminiscent capillarity**. The capillarity dependence of pore filling in nanopores/subnanopores as evidenced in Fig. 4 might appear as a surprising result. Yet, as will be shown in the remaining of this paper, such a scaling can be rationalized through simple thermodynamic arguments. Typically, the data shown in Fig. 4 suggest that a classical, macroscopic behavior remains meaningful at least in an effective way. While this result is rather unexpected for such ultraconfining pores, it is consistent with results from molecular simulation and classical DFT for simple pores which suggest that the scaling predicted from

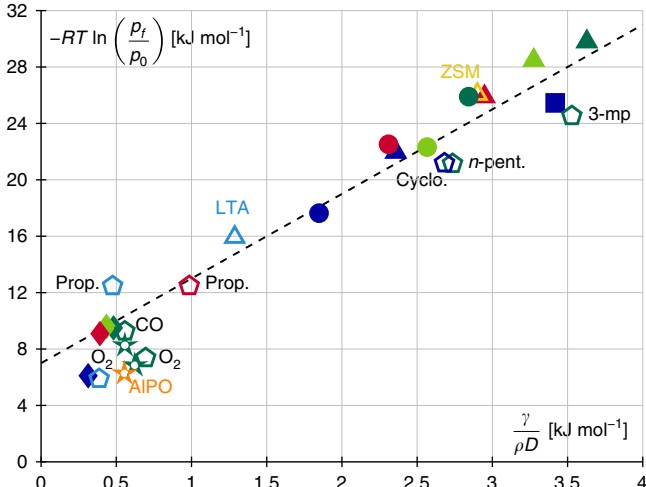

**Fig. 4** Master curve and generic scaling. Chemical potential $\mu \sim RT \ln(p_f/p_0)$ at which porosity filling occurs as a function of the capillary energy $\gamma/(\rho D)$ for different fluids in different zeosils for $\sigma/D > 1.4$. Filled symbols correspond to our molecular simulation data while open symbols correspond to available experimental data, with 2 additional matrices (light blue LTA, yellow ZSM-11, orange AlPO$_4$-5) and 6 additional fluids (pentagons except for Ar shown with stars) Argon (Ar), propylene (prop.), carbon monoxide (CO), oxygen (O$_2$), cyclohexane (cyclo.), 3-methylpentane (3-mp), n-pentane (n-pent). The dashed line denotes a linear relationship with $\mu \sim 6\gamma/\rho D$. The experimental data are taken from the following references: Ar/MFI at 77 and 87.3 K[43,44], n-C6/LTA[45], n-C6/ZSM-11[46], propylene/LTA et CHA[47], O$_2$ and CO/LTA and MFI[48], cyclo-C6/BEA[49], 2-MP/MFI[50], n-C5/MFI[51], Ar/AlPO$_4$-5[52].

Laplace equation remains appropriate at least qualitatively[24]. In fact, rather than a true capillarity effect, such capillarity dependence should be referred to reminiscent capillarity since pore filling in nanoporous and subnanoporous media exhibits specificity that departs from the well-established capillary regime. In particular, as explained in the introduction, for ultra-confining media, the absence upon adsorption of a well-defined gas/liquid interface within the porosity renders the concept of surface tension ambiguous. For such blurred interfaces, pore filling becomes reversible and continuous (second order transition) in stark contrast with the capillary-driven regime which corresponds to a discontinuous and irreversible process (first order transition). As a result, the pseudo-capillarity observed in Fig. 4 should rather be considered as reminiscent capillarity relevant to the strong surface to volume ratio in these systems with an associated surface tension that remains physical and meaningful down to pore sizes that are comparable with the granularity of the confined fluid.

As shown in what follows, such reminiscent capillarity can be predicted using simple classical thermodynamic modeling. Motivated by the simple capillarity scaling observed in Fig. 4, we attempt to predict vapor adsorption in zeosils using Derjaguin's formalism which allows describing both adsorption and condensation in porous media. In the framework of the Gibbs dividing surface concept, the grand potential $\Omega$ of a pore with a diameter $D$ and the fluid film of a thickness $t$ adsorbed at the vapor pressure $p_V$ and temperature $T$ writes[15,16]:

$$\Omega = -p_V V_V - p_L V_L + \gamma_{SL} A_{SL} + \gamma_{LV} A_{LV} + A_{LV} W_{SLV}(t) \quad (1)$$

where $p_V$, $p_L$, $V_V$, $V_L$, and $V_S$ are the pressure and volume of the vapor and adsorbed phases, respectively. $\gamma_{LV}$, $\gamma_{SL}$, $A_{LV}$, and $A_{SL}$ are the liquid/gas and liquid/solid surface tensions and surface areas, respectively. The interface potential $W_{SLV}(t)$ in the above equation, which allows describing adsorption at the solid surface,

accounts for the interaction between the liquid/solid and adsorbate/gas interfaces. $W_{SLV}(t)$ is linked to the disjoining pressure $\Pi(t) = -dW_{SLV}(t)/dt = p_V - p_L$ for planar interfaces[25,26]. Throughout this study, only bulk values are used for the liquid density and surface tension. However, the disjoining pressure can be seen as a correction to the bulk surface tensions due to interface coupling, i.e., $\Pi(t) = \partial(\gamma_{SL}(t) + \gamma_{LV}(t))/\partial t$, so that the approach above does account for confinement. In order to derive an expression for the surface potential $W_{SLV}(t)$, we write that it must verify the following condition: as $t$ becomes much larger than the characteristic interaction range $\xi$, the interactions between the gas/adsorbate and adsorbate/solid interfaces vanish—i.e., $W_{SLV}(t) \to 0$ for $t \to \infty$. On the other hand, for $t \to 0$, the adsorbate vanishes and the surface contribution in the grand potential given in Eq. (1) reduces to $\gamma_{SV} A_{SV}$ so that $W_{SLV}(0) \to S$ where $S$ is the spreading coefficient defined by[27]:

$$S = \gamma_{SV} - \gamma_{SL} - \gamma_{LV} \quad (2)$$

Among possible functions, $W_{SLV}(t) = S \exp(-t/\xi)$ verifies the above asymptotic limits and has been shown to allow reproducing adsorption of different fluids on various pore surfaces[28]. For a van der Waals fluid, $W_{SLV}(t) \sim H_{SLV}/t^2$ where $H_{SLV}$ is the so-called Hamaker constant—which is representative of the interaction strength between the solid/liquid and gas/liquid interfaces—is often proposed in the literature. In practice, such a power law scaling is valid for a finite film thickness $t$ only as it displays a non-physical divergence in the limit of vanishing films. To go beyond this specific van der Waals model and account for the non divergence of $W_{SLV}(t)$ when $t$ vanishes, several works suggest that a generic scaling $W_{SLV}(t) \sim \exp(-t/\xi)$ is more accurate and suitable to describe experimental/simulation data[16]. In capturing the so-called interface coupling, the disjoining pressure $\Pi(t)$ and the underlying surface potential $W_{SLV}(t)$ are effective parameters which also account for the microscopic details of nanoconfined fluids—such as the strong layering observed in classical DFT and molecular simulation—in a continuum behavior picture. In this sense, a generic and empirical exponentially decaying behavior for $W_{SLV}(t)$, with a prefactor and a lengscale corresponding to the strength and range of the interface coupling, is justified.

For a given gas pressure $p_V$, the stable solution predicted using Derjaguin's model is obtained by determining the minimum in the grand potential $\Omega(t)$—defined in Eq. (1)—upon varying $t$. For cylindrical mesopores, in agreement with previous works, the model is found to be in excellent agreement with the experimental data obtained for MCM-41 silica of a diameter $D = 4.7$ nm as shown in Fig. 5[29]. At low pressures, the only stable solution of the model corresponds to an adsorbed film of a thickness $t$. At the transition pressure $p_V^e$, the grand potential $\Omega(t)$ displays two minima which correspond to the same grand potential; These two solutions correspond to a configuration with an adsorbed film of finite thickness, i.e., $t < D/2$, coexisting with a configuration where the pore is completely filled with the liquid, i.e., $t = D/2$. At pressures above $p_V^e$, the film can remain stable in a metastable fashion until it collapses at the pressure where the corresponding grand potential minimum disappears. A single set of parameters in Derjaguin's model ($S = 0.069$ J m$^{-2}$ and $\xi = 0.24$ nm), fitted against a single experimental adsorption isotherm shown in Fig. 5a, allows reproducing, without any additional fitting and yet with a very good agreement, the experimental data obtained with other pore sizes (Fig. 5b). Note that both the parameters $S$ and $\xi$ are obtained from a single fit as there is only one combination that allows capturing quantitatively the adsorption (low pressure) and capillary condensation (high pressure) regimes. Clearly, the value $\xi \sim 0.24$ shows that the correction from the adsorbed film and the interface coupling is

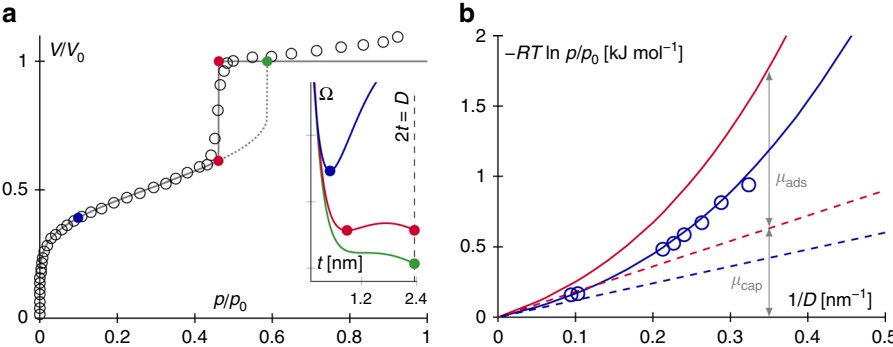

**Fig. 5** Thermodynamic modeling of adsorption in nanoporous materials. **a** Experimental $N_2$ adsorption isotherm at 77 K for a MCM-41 porous silica with cylindrical pores of a diameter $D = 4.7$ nm (circles). The lines are the predictions from Derjaguin's model as derived in the present work for a cylindrical geometry. The solid line represents the at-equilibrium data where the transition pressure is defined as the pressure for which the grand potential $\Omega$ for the filled pore equal that for the partially filled pore (an adsorbed film coexisting with the gas phase in the pore center). The dashed line corresponds to data with the condensation pressure defined as the pressure for which the partially filled pore becomes unstable. The insert shows the grand potential $\Omega(t)$ as a function of $t$ for the different regions of the adsorption isotherm. In the low pressure range, the stable solution (blue sphere) corresponds to the blue free energy profile: the solution corresponds to a minimum $t_e$ which corresponds to an adsorbed film. At the coexistence pressure, the two minima indicated by the red spheres correspond to the equality of the grand potentials, $\Omega(t_e) = \Omega(t = D/2)$. At the metastable condensation pressure (green sphere), the minimum corresponding to the partially filled pore disappears and the only stable solution corresponds to $t = D/2$. **b** Chemical potential shift $\mu_0 - \mu$ at which capillary filling in a pore of a diameter $D$ occurs. The shift $\mu_0 - \mu = -RT \ln(p/p_0)$ is defined as the difference between the chemical potential $\mu$ at condensation and the chemical potential at the bulk gas/liquid phase coexistence $\mu_0$ (note the negative sign which is used for convenience here). The red and blue solid lines are the Derjaguin model as derived in the present work for spherical and cylindrical pores, respectively (the dashed lines are the predictions from the Kelvin equation for these two geometries). The open circles are experimental data for MCM-41 porous silica with cylindrical pores having different diameters $D$[29]

far from being negligible for nanopores. In particular, in agreement with experimental data on regular porous silica (for which $D$ is simply a few times $\xi$), the correction due to the disjoining term—which decays over a typical lengthscale $\xi$—leads to quantitative departures from the conventional Kelvin equation. By comparing Derjaguin's model for the cylindrical and spherical geometries with the predictions of the corresponding Kelvin equation, Fig. 5b clearly shows that the chemical potential at pore filling is the sum of a capillary contribution, $\mu_{cap}$, given by the Kelvin equation derived for each pore geometry, and an adsorption contribution $\mu_{ads}$, which depends on the surface potential $W_{SLV}(t) = S \exp(-t/\xi)$ through the characteristic surface interaction range $\xi$ and spreading parameter $S$.

Figure 4 suggests that all the microporous structures considered in this work can be modeled using an effective spherical geometry as the slope ~6 of the master curve coincides with the expected value for spherical pores. The factor ~6 is far from being a trivial result as some of the zeolites considered here present pores closer to the cylindrical geometry. It is believed that, due to the severe confinement experienced by the fluid in these ultra-small pores, the confinement is equivalent in each of the 3 directions ($x$, $y$, $z$). In other words, when calculating the interaction field for a confined fluid, the direction along the pore axis is almost as confining as the other directions because of the vicinity of all zeolite atoms at the pore surface. Coming back to the thermodynamic model, for the spherical pore geometry, the vapor and liquid volumes expressed in terms of pore diameter $D$ and film thickness $t$ are $V_V = 1/6\pi(D - 2t)^3$ and $V_L = 1/6\pi[D^3 - (D - 2t)^3]$. The surface areas of the liquid/vapor and solid/liquid interfaces are, respectively, $A_{LV} = \pi(D - 2t)^2$ and $A_{SL} = \pi D^2$. For a given pore diameter $D$, the gas pressure $p_V^e(D)$ at which pore filling occurs can be determined by writing the following condition. The grand free energy $\Omega(t)$ of the configuration corresponding to the low density phase equals that of the filled configuration $\Omega(t = D/2) = -p_L V_L + \gamma_{SL} A_{SL}$ with $V_L = \pi D^3/6$. After a little algebra, one arrives at:

$$(D - 2t)(p_L - p_V^e(D)) + 6\gamma_{LV} + 6S \exp(-t/\xi) = 0 \quad (3)$$

Using the Gibbs–Duhem relation, i.e., $dp = \rho d\mu$, for both the adsorbed and gas phases, it can be shown that $p_L - p_V^e(D) = RT\rho_L \ln[p_V^e(D)/p_0]$ where $p_0$ is the bulk saturating vapor pressure while $R$ is the ideal gas constant (note that in deriving this relation we neglect the density of the gas phase before that of the adsorbed phase which is taken equal to the liquid phase density $\rho_L$). Upon inserting the latter expression in Eq. (3), one arrives at the filling pressure for a spherical pore of a diameter $D$ in which adsorption occurs prior to capillary filling:

$$RT \ln \frac{p_V^e(D)}{p_0} = -\frac{6\gamma_{LV}}{\rho_L(D - 2t)}\left[1 + \frac{S \exp(-t/\xi)}{\gamma_{LV}}\right] \quad (4)$$

The equilibrium configuration $\Omega(t)$ is a thermodynamic minimum, i.e., $d\Omega(t)/dt = 0$, which leads after derivation of Eq. (1) to the following expression: $S \exp(-t/\xi)(D - 2t + 4\xi) = -\rho_L RT \ln[p_V^e(D)/p_0](D - 2t)\xi - 4\gamma_{LV}\xi$. Upon inserting this expression into Eq. (4), it is straightforward to show that:

$$RT \ln \frac{p_V^e(D)}{p_0} = -\frac{6\gamma_{LV}}{\rho_L(D - 2t - 2\xi)} = -\frac{6\gamma_{LV}}{\rho_L D}\left[1 + \frac{2t + 2\xi}{D - 2t - 2\xi}\right] \quad (5)$$

This equation rigorously predicts the slope, $\mu \sim -6\gamma/\rho D$, observed in the linear master curve shown in Fig. 4. That capillarity remains relevant to pore filling in such small nanoporous materials is clearly an unexpected result. Yet, it is fully consistent with already available experimental, theoretical and molecular simulation data which all converge to show the following results. On the one hand, the pore condensation pressure is always found to be lower than the capillary condensation pressure predicted using simple surface to volume ratio arguments (Kelvin equation and any extension that includes ad hoc the adsorbed film thickness such as the well-known BJH method). On the other hand, such condensation pressures always follow the expected capillary scaling with a quantitative departure that vanishes as pores get large enough to recover the conventional macroscopic behavior. These results are fully consistent with the picture emerging from the present work with a chemical potential at pore

filling that includes both capillary and adsorption energy contributions.

**Adsorption contribution to filling transition**. Equation (5) also predicts in agreement with the same simulation and experimental data that the chemical potential at pore filling is shifted by an offset $K$ (i.e., $\mu \sim -6\gamma/\rho D + K$). Comparing the theoretical expression given in Eq. (4) and the linear scaling observed in Fig. 4 leads to $K = 6\gamma/\rho D \times 2(t + \xi)/(D - 2t - 2\xi) \sim 7 \pm 3 \text{ kJ mol}^{-1}$. Unexpectedly, this implies that the offset, which reflects adsorption effects, remains of the same order of magnitude for all the fluid/solid couples. While finding a similar adsorption energy $\sim K$ for the various adsorbate/zeolite couples considered here is intriguing, the value of $7 \pm 3 \text{ kJ mol}^{-1}$ can be rationalized as follows. Regardless of the system, physical adsorption occurs provided that the adsorption energy $K \sim \mu - \mu_0$ is at least larger than the thermal energy $RT$ (indeed, physisorption is not observed for $K <$ a few RT and much larger $K$ values correspond to chemisorption). For the range of fluids considered here, from $N_2$, Ar, and $O_2$ at low temperatures to organic fluids at room temperature ($RT \sim 0.6$ and $2.5 \text{ kJ mol}^{-1}$ for $T = 77$ and $300$ K, respectively), a few $RT$ leads to values in the range of $7 \pm 3 \text{ kJ mol}^{-1}$ as found in Fig. 4 where both experimental and molecular simulation data are plotted. The fact that the adsorption energy remains of the same order of magnitude for all systems implies that the relative adsorption contribution is limited compared to the capillary contribution for the points located in the right hand side of Fig. 4 (small pore limit). To assess the ability of the Derjaguin model derived here to predict this constant contribution through Eq. (5), we determined the film thickness $t$ of the adsorbed film at the onset of pore filling in the case of $N_2$ adsorption using the data for the different zeolites. As illustrated in Supplementary Fig. 7, this onset is revealed as a sudden transition in the filling behavior from Henry's (linear) regime at low pressure to an exponential dependence of $\log(n/n_0)$ according to the relative pressure at large pressures (see also Supplementary Discussion). The $N_2$ adsorbed amount at this transition point between these two regimes is modeled as a condensed film of thickness $t$ (more in detail, for a spherical geometry, the film thickness $t$ is readily extracted from the adsorbed amount $n$ at a given pressure $p$ as $2t/D = 1 - (1 - n/n_0)^{1/3}$ where $n_0$ is the maximum adsorbed amount). Figure 6 shows the critical film thickness—i.e., the thickness at the onset of pore filling—as a function of the pore diameter for $N_2$ adsorbed at 77 K in the different zeolites—both the film thicknesses determined from the simulated adsorption isotherms and from the Derjaguin model above are shown. We also compare the predictions of this model used for the cylindrical geometry with experimental data taken from the work by Kruk et al. for $N_2$ at 77 K in MCM-41 porous silica[29] (we note that the critical film thicknesses for the two geometries are equal and only depends on $D$ and $T$). As can be seen from the data in Fig. 5, the Derjaguin's model is in good agreement with the experimental and simulated data even when nanoporous and subnanoporous materials are considered. Interestingly, the reduced critical film thickness $2t(D)/D$ as a function of $D$ shows a non-monotonous behavior which can be rationalized as follows. On the one hand, for large pores, typically $D > 2$–3 nm, even if the critical thickness increases because the condensation pressure increases with $D$, it increases less rapidly than $D$ so that $2t/D$ is a decreasing function of $D$. On the other hand, for small pores, typically $D < 2$ nm, the filling pressure decreases so rapidly upon decreasing $D$ that the adsorbed film does not grow to significant values prior to pore filling (typically, for $D \lesssim$ nm, $t \lesssim 0.2$ nm). Interestingly, despite its derivation relying on macroscopic concepts that break down at the molecular scale, the Derjaguin model predicts very accurately the crossover between capillary condensation in large pores and

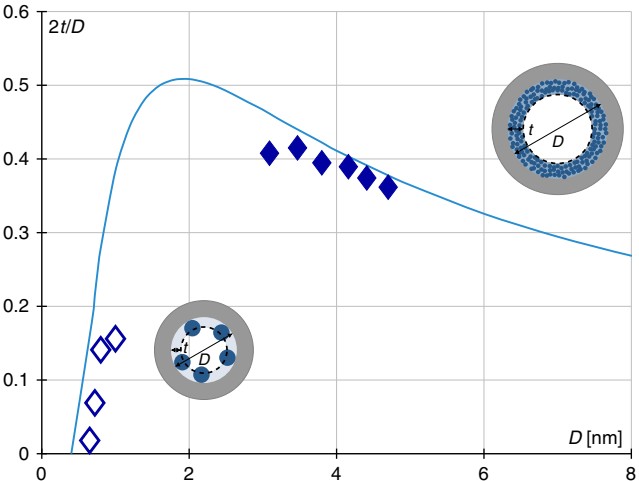

**Fig. 6** Critical fluid film thickness. Ratio $2t/D$ according to the pore diameter $D$ where $t$ is the thickness of a $N_2$ layer at a siliceous pore wall when the pore fills in. The line is a prediction from the minimization of the grand potential $\Omega$ taking into account the dependence of the Derjaguin disjoining pressure term with the thickness $t$. Symbols are experimental values for zeosils (open symbol, effective thickness) and for silica mesopores (filled symbol, condensed film thickness)[29]. The insets illustrate the ratio of the film thickness $t$ to the pore size $D$ for both the small (left) and large (right) nanopore domains. In both cases, the blue spheres denote the adsorbed fluid molecules while the gray and blue shaded areas indicate the adsorbed film and solid matrix, respectively

pore filling with vanishing adsorption in the ultra-small pores. Using the critical film thicknesses in Fig. 6, we estimated the value of the adsorption contribution to the chemical potential at pore filling, $K = 6\gamma/\rho D \times 2(t + \xi)/(D - 2t\, 2\xi)$ (we recall that this expression is obtained by writing that the second term in Eq. (5) is constant to be consistent with the behavior observed in Fig. 4). For the four different pores considered here, we found that $K \sim 7.7 \pm 3$ kJ mol$^{-1}$ which is in very good agreement with the offset, $K \sim 7 \pm 3$ kJ mol$^{-1}$, estimated from the linear scaling observed in Fig. 4.

**Discussion**
The agreement between the simulated/experimental data and the theoretical modeling supports the proposed idea of reminiscent capillarity in subnanoporous media. The underlying picture consists of a chemical potential at pore filling that can be subdivided into an adsorption energy and a capillary energy. For such ultra-small pores, these two terms are equally important but capillarity imposes a pore size dependence which is reminiscent of the macroscopic theory. Some remarks are in order regarding the use of the word "reminiscent" in "reminiscent capillarity". On the one hand, while our approach supports the idea that capillary concepts still apply in some effective and quantitative fashion, the fact that pore filling becomes reversible and continuous in very small pores (i.e., above the so-called capillary critical temperature $T_{cc}$) indicates that first-order capillary condensation does not occur. On the other hand, while capillary condensation does not apply stricto sensu to the situations considered here, the pore filling observed "does remind of capillary condensation" especially since the use of the corresponding quantitative parameters (surface tension, Laplace pressure, etc.) seems appropriate.

Far from being a trivial result, our genuine finding is very important by many practical and fundamental aspects. The unraveled reminiscent capillarity dependence goes well beyond a simple asymptotic limit as an important crossover between capillary condensation in large pores (first order phase transition)

and continuous and reversible filling in small pores (second order phase transition) still persists behind the apparent unifying capillary behavior established in the present work. Indeed, despite the unexpected robustness of capillarity as revealed in the present work, it is known that there is for a given temperature a critical pore diameter below which pore filling no longer proceeds through capillary condensation but corresponds to a continuous and reversible process. Equivalently, for a given pore size, there is a so-called capillary critical temperature above which capillary condensation is suppressed and replaced by continuous and reversible pore filling. Our findings are consistent with this picture and the apparent capillarity observed here suggests a reminiscent behavior of the physics in large pores rather than a simple mathematical extension. The applicability down to the molecular scale of macroscopic concepts such as those involved in capillarity (surface tension, Laplace pressure, etc.) is somewhat analog to the extension of hydrodynamics to fluids in nanoporous media. Indeed, while hydrodynamics in these ultraconfined environments must be corrected to account for novel phenomena such as slippage and memory effects, its underlying physics remains valid.

In any case, by bringing fundamental insights into the crossover between small and large nanopores (i.e., the conventional frontier between microporous solids $D < 2$ nm and mesoporous solids $D \sim 2$–50 nm), these findings help close—even if only partially—the gap between the classical/macroscopic thermodynamics of porous media and the nanophysics that has emerged with the advent of nanoporous materials. More in detail, while the filling/emptying kinetics for microporous and mesoporous solids are expected to be different, the idea of underlying capillary concepts that remain meaningful even in ultraconfined environments allows reconciling these two regimes. In the light of these results, the nanoscale appears as a particular lengthscale where different physical effects compete: adsorption, confinement, surface interactions, etc. From a practical viewpoint, while conventional techniques are usually valid for a given fluid in a specific type of porous solids, our results also pave the way for extended characterization techniques that would apply to any system. However, by many aspects, the simple model used to account for such reminiscent capillarity must be improved as it remains at this stage limited to homogeneous porous solids. For instance, the theoretical approach reported here does not apply to solids exhibiting surface chemistry heterogeneity where adsorption sites lead to a complex adsorption contribution in the chemical potential at pore filling. This includes for instance processes involving adsorption of polar molecules on very strong adsorption sites but also adsorption in cationic zeolites. Moreover, as in the case of porous solids with important morphological (pore shape) disorder, we do not expect the simple approach used in the present work to apply for highly disordered materials although the concept of capillarity and surface free energy minimization with respect to the free energy volume should remain key ingredients. Finally, in recent years, an increasing amount of effort has been devoted to the coupling of adsorption and deformation in nanoporous solids. The thermodynamic formalism to capture such effects is intrinsically more complex than for non-compliant materials (typically, one has to use a hybrid ensemble with a thermodynamic potential that includes both the grand free energy of the fluid and the free energy or free enthalpy of the host solid). Yet, as far as the fluid contribution is concerned, we expect the results reported in the present work to remain meaningful.

## Methods

**Adsorption experiments**. Nitrogen adsorption/desorption isotherms were measured using a Micromeritics ASAP 2420 apparatus. Prior to the adsorption measurements, the calcined samples were out-gassed at 300 °C overnight under

vacuum. For the other fluids considered in this work, dynamic adsorption measurements were performed under different atmospheres of volatile organic compounds (n-hexane, p-xylene, acetone) and controlled values of relative pressure $p/p_0 = 0.5$ ($p$ is the pressure and $p_0$ the saturation vapor pressure at a given temperature $T$ of the considered organic compound) using a thermogravimetric balance Setaram TG92 instrument[30]. The experiments were done under flow. The relative pressure $p/p_0 = 0.5$ was obtained by setting the pressures of auxiliary gas and carrier gas to 1.5 bar at the inlet of the oven and controlled by measuring the gas flow rate at the outlet of the oven. The gas flow rate was found to be stable (114 ml.min$^{-1}$). Prior to the adsorption experiment, a preliminary activation phase was accomplished which consisted in heating up the zeosil to 350 °C with the aim to remove all adsorbate traces. Subsequently, the sample was cooled back to T = 25 °C and the organic compound was then introduced in the system to saturate the zeosil. The adsorbed amount was reported every 20 s. Experiments were performed on 100 mg of zeosil.

**Computational models**. All investigated zeolite structures (Supplementary Fig. 1) were simulated under their purely silicate form and were maintained rigid during the simulation, with framework atoms fixed to their crystallographic positions. For CHA, STT, and BETAPA type zeolites the atomic positions were taken from the IZA database. For the zeolite beta, whose crystal structure is formed by an intergrowth of two crystallographic forms, designed, respectively, A and B, we have focused uniquely on the A polymorph in our simulation work. The silicalite-1 is known to exist in three distinct forms: a monoclinic one with Pnma space group and two orthorhombic forms with Pnma and P212121 space groups, designed respectively as MONO, ORTHO, and PARA. silicalite-1 is usually synthesized in the ORTHO form and passes into the low temperature MONO system after calcination. Further, this low temperature form (MONO) experiences a reversible phase transition into the ORTHO structure at about 350 K, as well as under adsorption of various molecules. Finally, upon adsorption of certain adsorbates such as nitrogen at 77 K or p-xylene the ORTHO structure moves into the PARA one. Thus, in order to reproduce in a basic manner the experimental isotherms, we have considered separately the low pressure (ORTHO) and the high pressure (PARA) systems as previously done by Snurr et al.[31]. The atomic positions for the ORTHO silicalite-1 structure were taken from the IZA database[32], whereas those for the PARA silicalite-1 were taken from van Koningsveld et al.[33].

The investigated molecules are described through a "united atom" model that has been employed successfully for the investigation of their adsorption behavior in zeolites[34] and MOFs[35]. In this model, each –CH$_x$– (with $0 \geq x \leq 3$) and (=O) group is treated as a single interaction site. Such "united atoms" are connected by bonds maintained at fixed distances. In addition, several interaction sites bear a partial charge, contributing to the interaction energy through the Coulombic term. The nitrogen molecule is described via an explicit model: each nitrogen atom of the rigid molecule constitutes a single interaction center bearing a negative partial charge. In order to compensate the negative charges on the nitrogen atoms, there is a positive partial charge bearing no-interacting site in the middle of the nitrogen-nitrogen bond. Such three sites model allows reproducing the experimentally measured quadrupole moment of the nitrogen molecule. The intermolecular interactions between the adsorbate molecules were modeled using a sum of repulsion-dispersion potential term expressed as the Lennard–Jones interaction and the Coulombic interaction. The cross LJ terms were calculated applying the Lorentz–Berthelot combination rules. The bond distances, the partial charges and the interatomic potential parameters for all investigated molecules are summarized in Supplementary Table 3. Within the frame of the selected models, the nitrogen and p-xylene molecules are considered as rigid; therefore no intramolecular interactions are taken into account. While the intramolecular interactions for the acetone molecules are described solely via an harmonic bending term, for the n-hexane an additional dihedral torsion angle term is considered expressed by a cosine series potential. The parameters corresponding to those terms have been taken from the Transferable Potential for the Phase Equilibrium (TraPPE) forcefield, respectively, for n-hexane[36], p-xylene[37], acetone[38], and nitrogen[39] (initially fitted to reproduce the liquid/vapor coexistence curves of various fluid molecules and summarized in Supplementary Table 3).

The absolute adsorption isotherms of n-hexane, acetone, p-xylene at 298 K and nitrogen at 77 K were computed in each zeosil using the Monte Carlo simulation within the Grand Canonical ensemble implemented within the code MCCCS Towhee[40]. These simulations consisted of evaluating the average number of adsorbate molecules whose chemical potential equals those of the bulk phase for given chemical potential and temperature. The chemical potential values were calculated by the test particle Widom insertion method from the NpT ensemble Monte Carlo simulation. The conventional scheme of the GCMC simulation for flexible, long chain molecules is expensive in computational time as the fraction of successful insertion moves is too low. Moreover, it does not fully explore the conformation part of the configuration space and thus does not allow achieving a proper distribution of bending and dihedral angles. The configurational-biased algorithm overcomes such shortcomings by sampling more efficiently the configuration space[23]. We have applied the coupled – decoupled biased selection scheme developed by Martin and Siepmann, performing a coupled biased selection for Lennard-Jones and torsion angles selection steps, while decoupling the angle bending energy into split biased selections. The detail description of the particular

algorithm employed in our simulation can be found in Martin and Siepmann[41,42]. The structures of considered zeosils were treated as rigid and the periodic conditions were applied. A typical Monte Carlo run consisted of $3 \times 10^6$ steps. Each step corresponded to a single MC move, including a center of mass translation, center of mass rotation, insertion of a new molecule, deletion of a randomly selected existing molecule, partial or complete regrowth of the adsorbate. The Ewald summation technique was used in the calculation of the long-range electrostatic interactions.

## Data availability

The datasets generated during and/or analyzed during the current study are available from the corresponding authors on reasonable request. All simulations were performed using the software TOWHEE: MCCCS Towhee-Version 7.0.6 (July 27, 2013).

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

## Acknowledgements
I.D. thanks the Pôle HPC and Equipex Equip@Meso at the University of Strasbourg. This work was supported by the French Research Agency (ANR TAMTAM 15-CE08-0008 and LyStEn 15-CE06-0006).

## Author contributions
I.D. and B.C. designed the work. I.D. performed and analyzed the molecular simulations while T.J.D. carried out the adsorption experiments. B.C. developed the theoretical model with the help of C.P. B.C wrote the paper with input from all the authors.

## Competing interests
The authors declare no competing interests.
