## [Peer Review File · Nature Communications]

Reviewers' comments:

Reviewer #1 (Remarks to the Author):

Based on experimental and atomistic simulation data on a variety of model porous materials and adsorbates, this article shows that the filling of nanometric or subnanometric pores can still be described effectively by macroscopic capillarity concepts. This is a very interesting result, which I think will influence thinking in the field and pave the way for investigations on more complex materials. I therefore recommend publication of this article in Nature Communication after the following concerns have been considered.

1. I am somewhat surprised by the contrast between the continuum behavior of fluids in nanometric confinement observed here, and the strong layering effects observed experimentally and numerically (e.g. with classical DFT or molecular dynamics) in other contexts (both for structure and dynamics, including studies of capillary filling). I think this would deserve at least a short discussion, in particular for the choice of the interfacial potential W_{SLV} .

2. At the same time, when the authors mention that the validity of capillarity at the nanometer scale is surprising, they could mention such validity has been observed in other fields, e.g. hydrodynamics.

3. I also feel the pore diameter D and fluid molecular size σ are key quantities, so that while the authors explain how D was estimated in the caption of Fig. 2, they should also explain how σ was computed. Similarly, I didn't find a rigorous definition of the pore filling pressure, while this is one of the key quantities entering into the main relation of the article.

4. Also, when the authors refer to density and surface tension, I think they should clarify that they are using the macroscopic values. Indeed, this is not trivial and this is key to the overall message of the article.

5. In Fig. 4, the authors show that data for pore filling of various adsorbates in various porous materials fall onto a master curve. Yet I was wondering about the use of a logarithmic axis for the filling pressure p_f . I think it could be interesting to mention what is the typical difference between the p_f predicted by their master curve and the measured p_f , and to discuss whether this is acceptable for applications.

Minor:

6. I am not fully convinced by the term *reminiscent*, which at least for me conveys the idea of memory. I would rather have used something like "effective capillarity", but I totally understand if this is not to the taste of the authors.

7. Page 11, Figure 5, line 3, there is a doubled "the"

Reviewer #2 (Remarks to the Author):

This is an interesting paper, demonstrating that capillarity in zeolites can be described by a conveniently simple theory that adds two energies: adsorption energy and capillary energy.

I have one question and a number of remarks.

The question: the authors find a contribution of adsorption that is 7 plm 3 kJ/mol. Is this adsorption energy in line from what is known from literature sources?

Remarks:

- I think the word 'reminiscent' does not convey the meaning the authors think it does. Reminiscent means: 'reminds me of'. The others I think mean something like 'persistent' (but also that word does not entirely have the intended meaning I think).
- The structure of the paper can be improved somewhat. When the authors show figure 4, 'the master curve', they call it 'a striking result', 'which questions existing frameworks'. In the end we however find out that it can be captured by classical thermodynamic modeling, and comes down to this additivity of the different energies. We would be helped when the authors directly at figure 4 point out the main characteristics of the relationship found, which are the system obeying the Kelvin equation with an exponent of 6, with an added offset. They can then proceed with applying Derjaguin's formalism for a spherical space.
- Finally, I am not sure of part of the discussion. It seems the authors are expecting that the reversibility or non-reversibility of the filling process should have a bearing on the thermodynamic energy of the final state. I think it doesn't matter whether we get there by capillary condensation or surface diffusion.

Reviewer #3 (Remarks to the Author):

This work presents very impressive results, which are absolutely unexpected to me, and (I suspect) to most people in our field. I am surprised that the scaling shown by authors in Figure 4 has not been discovered before.

The results are novel and groundbreaking for communities working in adsorption or confined fluid. The diversity and the number of system studies are impressive. The paper is clearly and interestingly written, and in opinion it is certainly worth publishing in Nature Communications. Nevertheless, I have a number of minor comment, which would be nice to address prior to acceptance.

Main comment: on Fig. 4 the temperature is missing. Given the amount of work the authors already invested in this study, I do not want to run additional simulations, but maybe the authors could find literature data for gas adsorption in those zeolites, when adsorption of the same gas is done at different temperatures, and plot the points on Fig. 4? A simple example would be argon at 77K and 87K, although a broader temperature interval would be better.

Other comments:

- How reliable is Widom insertion at these tiny confinement and high density?
- The discussion about the bulk density and equilibration is not clear
- In the recent years a lot of work (including some of these authors) has been done on materials which deform upon adsorption. It would be interesting if the author at least speculate (in the discussion section) how adsorption-induced deformation could affect their results.
- In my opinion MFI zeolites have cylindrical pores. It would be nice if authors commented on the spherical pore model applied to them.
- In Eq. 3, when $D < 1$ nm and $\lambda_{xi} = 0.24$ nm, it is not a small correction, and it is worth a more detailed discussion (possibly some plot)
- Where does the fitting of λ_{xi} come from: is it from some reference materials, or is done from the same adsorption isotherm?
- Code availability: version of Towhee is not mentioned. Moreover, since it is a widely used open-source code, it would be helpful for a potential reader, if the input files and initial coordinates would be provided

SI:

Supplementary Figure 3: it would be better to represent the pressure in the log scale, also the number notation on the scale should be consistent with the article: 0.2 instead of 0,2, etc.

Typos:

"data where the the"

"experimental, theoretical and molecular simulations which all " a noun is missing

* * * * *

Reviewer #1

Based on experimental and atomistic simulation data on a variety of model porous materials and adsorbates, this article shows that the filling of nanometric or subnanometric pores can still be described effectively by macroscopic capillarity concepts. This is a very interesting result, which I think will influence thinking in the field and pave the way for investigations on more complex materials. I therefore recommend publication of this article in Nature Communication after the following concerns have been considered.

Comment 1. *I am somewhat surprised by the contrast between the continuum behavior of fluids in nanometric confinement observed here, and the strong layering effects observed experimentally and numerically (e.g. with classical DFT or molecular dynamics) in other contexts (both for structure and dynamics, including studies of capillary filling). I think this would deserve at least a short discussion, in particular for the choice of the interfacial potential W_{SLV} .*

Reply 1. We thank the reviewer for this interesting comment. In fact, the surface potential $W_{SLV}(t)$ and the corresponding disjoining pressure $\Pi(t)$ are effective parameters which do include – in a non explicit fashion – all the microscopic details into a continuum approach. In so doing, these contributions capture the physical coupling between two interfaces – liquid/solid and liquid/gas – but also molecular effects which cannot be described otherwise in such a macroscopic model. In order to address this important comment, we have added the following discussion in the revised version of the manuscript (Page 11): “In capturing the so-called interface coupling, the disjoining pressure $\Pi(t)$ and the underlying surface potential $W_{SLV}(t)$ are effective parameters which also account for the microscopic details of nanoconfined fluids -- such as the strong layering observed in classical DFT and molecular simulation - - in a continuum behavior picture. In this sense, a generic and empirical exponentially decaying behavior for $W_{SLV}(t)$, with a prefactor and a lengthscale corresponding to the strength and range of the interface coupling, is justified.”

Comment 2. *At the same time, when the authors mention that the validity of capillarity at the nanometer scale is surprising, they could mention such validity has been observed in other fields, e.g. hydrodynamics.*

Reply 2. We agree with the reviewer. In order to address this comment, we have added the following discussion in the revised version of the discussion (Page 17): “Our findings are consistent with this picture and the apparent capillarity observed here suggests a reminiscent behavior of the physics in large pores rather than a simple mathematical extension. The applicability down to the molecular scale of macroscopic concepts such as those involved in capillarity (surface tension, Laplace pressure, etc.) is somewhat analogous to the extension of hydrodynamics to fluids in nanoporous media. Indeed, while hydrodynamics in these ultraconfined environments must be corrected to account for novel phenomena such as slippage and memory effects, its underlying physics remains valid.”

Comment 3. *I also feel the pore diameter D and fluid molecular size σ are key quantities, so that while the authors explain how D was estimated in the caption of Fig. 2, they should also explain how σ was computed. Similarly, I didn't find a rigorous definition of the pore filling pressure, while this is one of the key quantities entering into the main relation of the*

article.

Reply 3. We thank the reviewer for bringing these issues to our attention. In order to address these points, we have added the following sentences in our revised manuscript:

- Page 5: “For each fluid, the molecular size σ was taken as the kinetic diameter which is estimated to match the second virial coefficient. This choice is particularly suitable as it applies both to fluids in gas-like and liquid-like states.”
- Page 7: “For each system, the filling pressure p_f/p_0 was estimated as the pressure at which pores get half-filled. This definition is a robust way to characterize the position of the sharp variation of the pore content revealed in semi-log plot. While other definitions could be used, they would not change the outcome of our discussion below.”

Comment 4. *Also, when the authors refer to density and surface tension, I think they should clarify that they are using the macroscopic values. Indeed, this is not trivial and this is key to the overall message of the article.*

Reply 4. We agree with the reviewer that this is an important point. Indeed, only bulk values are used throughout this study for the liquid density and surface tension. However, we recall that the disjoining pressure is in fact a correction to the different surface tensions under confinement (i.e. when the different interfaces see each other). More in detail, the disjoining pressure describes the change in the two surface tensions as the distance t separating the two corresponding interfaces increases: $\Pi(t) = \partial[\gamma_{SL}(t) + \gamma_{LV}(t)]/\partial t$. This leads to $\gamma_{SL}(t) + \gamma_{LV}(t) = \gamma_{SL}(\infty) + \gamma_{LV}(\infty) + \int \Pi(t) dt$. Consequently, while only bulk values are used (which makes our approach particularly simple and applicable), confinement effects on surface tensions are properly included through the disjoining pressure and the underlying surface potential. In order to clarify this issue, we have added the following sentences in the revised manuscript (Page 11): “Throughout this study, only bulk values are used for the liquid density and surface tensions. However, the disjoining pressure can be seen as a correction to the bulk surface tensions due to interface coupling, i.e. $\Pi(t) = \partial(\gamma_{SL}(t) + \gamma_{LV}(t))/\partial t$, so that the approach above does account for confinement.”

Comment 5. *In Fig. 4, the authors show that data for pore filling of various adsorbates in various porous materials fall onto a master curve. Yet I was wondering about the use of a logarithmic axis for the filling pressure p_f . I think it could be interesting to mention what is the typical difference between the p_f predicted by their master curve and the measured p_f , and to discuss whether this is acceptable for applications.*

Reply 5. This is indeed an important comment as the log scale tends to compress all the data. We agree with the reviewer that this calls for further discussion which has been added in the revised version of the manuscript (Page 8): “The use of the chemical potential shift $\mu - \mu_0$ instead of the filling pressure p_f/p_0 is justified by the fact that the former is the appropriate driving force for capillary condensation/density change (since the chemical potential is the conjugated variable of the number of molecules in the grand free energy). However, in the meantime, the use of a log scale $\mu \sim RT \ln p$ introduces a non-negligible uncertainty when using the master curve shown in Fig. 4 to predict filling pressures. At worse, the relative uncertainty over the chemical potential at pore filling is about 25% which leads to the same relative uncertainty for the log of the filling pressure. While this seems reasonable given the broad applicability of the master curve provided here, this uncertainty should be included when rigorously estimating the filling pressure for a

specific example.”

Comment 6 (minor). *I am not fully convinced by the term *reminiscent*, which at least for me conveys the idea of memory. I would rather have used something like "effective capillarity", but I totally understand if this is not to the taste of the authors.*

Reply 6. We thank the reviewer for this insightful comment (which was also raised by reviewer #2). We put a lot of thought into the word “reminiscent” and also considered alternatives such as “effective” and “persistent”. Let us start with the adjective “persistent”. While our approach supports the idea that capillary concepts still apply in some effective and quantitative fashion, the fact that pore filling becomes reversible and continuous in very small pores (i.e. above the so-called capillary critical temperature T_{cc}) indicates that a first-order transition such as capillary condensation does not occur. In that sense, while “persistent” is indeed a word that applies to some extent to the findings reported here, we feel that its use alone would be misleading. As for “effective”, we fully agree with the reviewer that this word is appropriate but we feel that it is also misleading as it is often used in a different context (like an effective parameter taking into account other effects such as described earlier for the surface potential W_{SLV}). Therefore, while we agree that “reminiscent” might not be perfect, we think this is the word that best captures our findings: While capillary condensation does not apply *stricto sensu*, the pore filling observed here “does remind of capillary condensation” including through the use of the corresponding quantitative parameters (surface tension, Laplace pressure, etc.). We understand from the reviewer’s comment (which, again, was also raised by reviewer #2) that this issue must be better discussed. In order to address this comment, we have added the following discussion in the revised version of the manuscript (Page 16): “Some remarks are in order regarding the use of the word “reminiscent” in “reminiscent capillarity”. On the one hand, while our approach supports the idea that capillary concepts still apply in some effective and quantitative fashion, the fact that pore filling becomes reversible and continuous in very small pores (i.e. above the so-called capillary critical temperature T_{cc}) indicates that first-order capillary condensation does not occur. On the other hand, while capillary condensation does not apply *stricto sensu* to the situations considered here, the pore filling observed “does remind of capillary condensation” especially since the use of the corresponding quantitative parameters (surface tension, Laplace pressure, etc.) seems appropriate.”

Comment 7 (minor). *Page 11, Figure 5, line 3, there is a doubled "the"*

Reply 7. We thank the reviewer for bringing to our attention this typo which has been corrected in the revised version of the manuscript.

* * * * *

Reviewer #2

This is an interesting paper, demonstrating that capillarity in zeolites can be described by a conveniently simple theory that adds two energies: adsorption energy and capillary energy. I have one question and a number of remarks.

Comment 1. *The authors find a contribution of adsorption that is 7 plm 3 kJ/mol. Is this*

adsorption energy in line from what is known from literature sources?

Reply 1. This is indeed an important result of the present study that the typical adsorption energy falls in a range close to 7 kJ/mol. Given the diversity of the adsorbates/zeolites considered here, this result – which seems to be verified both using experimental and molecular simulation data – is particularly intriguing. The value of 7 +/- 3 kJ/mol can be rationalized as follows. Regardless of the adsorbate/adsorbent chosen, physical adsorption occurs provided that the adsorption energy $\mu - \mu_0 = K$ is at least larger than the thermal energy $k_B T$. For the range of fluids considered here, from N₂, Ar, and O₂ at low temperatures to organic fluids at room temperature ($k_B T \sim 0.6$ and 2.5 kJ/mol for $T = 77$ K and 300 K, respectively), a few $k_B T$ leads to values in the range of 7 +/- 3 kJ/mol as found in Fig. 4 where both experimental and molecular simulation data are plotted (much larger K would correspond to chemisorption). This is a very important point which we have further discussed in the revised version of the manuscript (Page 15): “While finding a similar adsorption energy $\sim K$ for the various adsorbate/zeolite couples considered here is intriguing, the value of ~ 3 kJ/mol can be rationalized as follows. Regardless of the system, physical adsorption occurs provided that the adsorption energy $K \sim \mu - \mu_0$ is at least larger than the thermal energy RT (indeed, physisorption is not observed for $K < a \text{ few } RT$ and much larger K values correspond to chemisorption). For the range of fluids considered here, from N₂, Ar, and O₂ at low temperatures to organic fluids at room temperature ($RT \sim 0.6$ and 2.5 kJ/mol for $T = 77$ K and 300 K, respectively), a few RT leads to values in the range of ~ 3 kJ/mol as found in Fig. 4 where both experimental and molecular simulation data are plotted.”

Comment 2. *I think the word 'reminiscent' does not convey the meaning the authors think it does. Reminiscent means: 'reminds me of'. The others I think mean something like 'persistent' (but also that word does not entirely have the intended meaning I think).*

Reply 2. We thank the reviewer for this insightful comment (which was also raised by Reviewer #1). This issue was already addressed in our reply to comment 6 from Reviewer #1.

Comment 3. *The structure of the paper can be improved somewhat. When the authors show figure 4, 'the master curve', they call it 'a striking result', 'which questions existing frameworks'. In the end we however find out that it can be captured by classical thermodynamic modeling, and comes down to this additivity of the different energies. We would be helped when the authors directly at figure 4 point out the main characteristics of the relationship found, which are the system obeying the Kelvin equation with an exponent of 6, with an added offset. They can then proceed with applying Derjaguin's formalism for a spherical space.*

Reply 3. We thank the reviewer for these suggestions which have been included in the revised manuscript. More in detail, in line with these suggestions, we have rewritten the following discussion to tone down our claims and make clearer that our results can be rationalized as shown in the remaining of the paper (Page 10): “The capillarity dependence of pore filling in nanopores/subnanopores as evidenced in Fig. 4 might appear as a striking result. Yet, as will be shown in the remaining of this paper, such a scaling can be rationalized through simple thermodynamic arguments. Typically, the data shown in Fig. 4 suggest that a classical, macroscopic behavior remains meaningful at least in an effective way. While this result is rather unexpected for such ultraconfining pores, it is consistent with results from molecular simulation and classical DFT for simple pores which suggest that the scaling predicted from Laplace equation remains appropriate at least qualitatively \cite{Ravikovitch-1998}.”

Comment 4. *Finally, I am not sure of part of the discussion. It seems the authors are expecting that the reversibility or non-reversibility of the filling process should have a bearing on the thermodynamic energy of the final state. I think it doesn't matter whether we get there by capillary condensation or surface diffusion.*

Reply 4. We think that there is some confusion here which should be clarified. We do not think that the underlying dynamics should affect the thermodynamic energy of the final states (as thermodynamics does not depend on the dynamical path followed by the system). In fact, it is meant here that our finding allows bridging the gap between the thermodynamics for large pores (mesoporous solids and solids with even larger pores) and for very small pores (microporous solids). We have clarified this issue by modifying the following discussion (Page 18): “In any case, by bringing fundamental insights into the crossover between small and large nanopores (i.e. the conventional frontier between microporous solids $D < 2$ nm and mesoporous solids $D \sim 2 - 50$ nm), these novel findings help close -- even if only partially -- the gap between the classical/macroscopic thermodynamics of porous media and the nanophysics that has emerged with the advent of nanoporous materials. More in detail, while the filling/emptying kinetics for microporous and mesoporous solids are expected to be different, the idea of underlying capillary concepts that remain meaningful even in ultraconfined environments allows reconciling these two regimes.”

* * * * *

Reviewer #3

This work presents very impressive results, which are absolutely unexpected to me, and (I suspect) to most people in our field. I am surprised that the scaling shown by authors in Figure 4 has not been discovered before. The results are novel and groundbreaking for communities working in adsorption or confined fluid. The diversity and the number of system studies are impressive. The paper is clearly and interestingly written, and in opinion it is certainly worth publishing in Nature Communications. Nevertheless, I have a number of minor comment, which would be nice to address prior to acceptance.

Comment 1 (minor). *On Fig. 4 the temperature is missing. Given the amount of work the authors already invested in this study, I do not want to run additional simulations, but maybe the authors could find literature data for gas adsorption in those zeolites, when adsorption of the same gas is done at different temperatures, and plot the points on Fig. 4? A simple example would be argon at 77K and 87K, although a broader temperature interval would be better.*

Reply 1. We agree with the reviewer that the effect of temperature is indeed a very important point that allows further validating our approach. We would like to emphasize that such a consistency check was already included in our first submission since our master curve in Fig. 4 already included several experimental and simulation points at low temperature (typically, for instance, O₂, N₂ at 77 K). More importantly, following the reviewer's comment, we decided to extend our search for available experimental data to further support our finding (which is not an easy task since most experimental data do not provide data in a log scale at pressures low enough to determine unambiguously the filling pressure). More in detail, in line with the reviewer's suggestion, we have added two data points for Ar at 77 K and 87.3 K in MFI. We have also added a point for Ar at 77 K in a non-silica zeolite which further extends the validity of our master curve. These three data points are shown in the revised version of Fig. 4 as the

dark green open stars (Ar 77 K and 87.3 K in MFI) and orange open star (Ar 77 K in $\text{AlPO}_4\text{-5}$). As can be seen from the revised Fig. 4, these additional data do follow the scaling shown by the previously reported data. We understand from the reviewer's comment that these points – i.e. temperature effect and extension to non-silica zeotype – should be better discussed. To do so, in addition to the revised version of Fig. 4, we have added the following sentences (Page 8): “In particular, the data used to establish such a correspondence between chemical potential at pore filling and capillary energy correspond to different fluid molecule shapes -- small regular versus chain molecules -- with chemical structures being either polar or apolar. Moreover, both the simulated and experimental data used to build the plot in Fig. 4 were obtained for fluids at different temperatures (typically, at room temperature and/or at temperature of boiling nitrogen). The fact that all these data follow the same trend described by the simple scaling above further supports the underlying theoretical picture. As a further validation, the data shown in Fig. 4 also includes a data point obtained for a non-silica zeotype – Ar at 77 K in the aluminophosphate $\text{AlPO}_4\text{-5}$ -- which suggests that this scaling also applies to other members of this broad porous solid family.”

Comment 2 (minor). *How reliable is Widom insertion at these tiny confinement and high density?*

Reply 2. The reviewer is correct in suggesting that the Widom insertion technique might be unreliable for confined fluids (it is indeed very complicated especially for some of the complex fluids considered here). However, in the context of the present study, the Widom insertion technique was only performed for the bulk fluids and not for the confined fluids. Indeed, since we are only interested here in determining the relationship between the bulk chemical potential and pressure $\mu(P)$, such calculations were carried out for the bulk fluids in the framework of NPT Monte Carlo simulations (i.e. at constant number of molecules N , pressure P , and temperature T). In order to clarify this point, we have added/modified the following sentences in the revised version of the manuscript (Page 6): “Such simulations are efficient for bulk phases as considered here (since the relationship $\mu(P)$ has to be determined for the bulk fluids only).”

Comment 3 (minor). *The discussion about the bulk density and equilibration is not clear.*

Reply 3. In order to address this comment, we have added/rewritten the following sentences (Page 6): “As for the adsorption simulations using the GCMC algorithm, especially for complex fluids such as some of the fluids considered here, it is known that proper equilibration to reach the physical density of the confined fluid suffers from technical limitations. More in detail, due to the intrinsic difficulty in inserting molecules into the small pores of nanoporous solids, GCMC can be very slowly converging to reach the equilibrium density. To circumvent poor sampling efficiency and guarantee convergence towards equilibrium, our GCMC molecular simulations were performed using the Configurational-Bias algorithm \cite{siepmann-molphys-1992}.”

Comment 4 (minor). *In the recent years a lot of work (including some of these authors) has been done on materials which deform upon adsorption. It would be interesting if the author at least speculate (in the discussion section) how adsorption-induced deformation could affect their results.*

Reply 4. This is indeed an interesting issue which is now discussed in the revised discussion of our manuscript (Page 18): “Finally, in recent years, an increasing amount of effort has been devoted to the coupling of adsorption and deformation in nanoporous solids. The thermodynamic

formalism to capture such effects is intrinsically more complex than for non-compliant materials (typically, one has to use a hybrid ensemble with a thermodynamic potential that includes both the grand free energy of the fluid and the free energy or free enthalpy of the host solid). Yet, as far as the fluid contribution is concerned, we expect the results reported in the present work to remain meaningful.”

Comment 5 (minor). *In my opinion MFI zeolites have cylindrical pores. It would be nice if authors commented on the spherical pore model applied to them.*

Reply 5. This is indeed an important point which remains to be fully understood. In order to clarify this issue, we have added the following sentences in the revised version of the manuscript (Page 12): “The factor $\xi \sim 6$ is far from being a trivial result as some of the zeolites considered here present pores closer to the cylindrical geometry. It is believed that, due to the severe confinement experienced by the fluid in these ultra-small pores, the confinement is equivalent in each of the 3 directions (x, y, z). In other words, when calculating the interaction field for a confined fluid, the direction along the pore axis is almost as confining as the other directions because of the vicinity of all zeolite atoms at the pore surface.”

Comment 6 (minor). *In Eq. 3, when $D < 1$ nm and $\xi = 0.24$ nm, it is not a small correction, and it is worth a more detailed discussion (possibly some plot)*

Reply 6. We agree that this point should be better discussed. In order to address this comment, we have modified/rewritten the following discussion in the revised version of the manuscript (Page 12): “Clearly, the value $\xi \sim 0.24$ shows that the correction from the adsorbed film and the interface coupling is far from being negligible for nanopores. In particular, in agreement with experimental data on regular porous silica (for which D is simply a few times ξ), the correction due to the disjoining term - which decays over a typical lengthscale ξ - leads to quantitative departures from the conventional Kelvin equation. By comparing Derjaguin’s model for the cylindrical and spherical geometries with the predictions of the corresponding Kelvin equation, Fig. 5B clearly shows that the chemical potential at pore filling is the sum of a capillary contribution, μ_{cap} , given by the Kelvin equation derived for each pore geometry, and an adsorption contribution μ_{ads} , which depends on the surface potential $W_{\text{SLV}}(t) = S \exp(-t/\xi)$ through the characteristic surface interaction range ξ and spreading parameter S .”

Comment 7 (minor). *Where does the fitting of ξ come from: is it from some reference materials, or is done from the same adsorption isotherm?*

Reply 7. As correctly assumed by the reviewer, ξ is fitted from the adsorption isotherm shown in Fig. 5A. We understand from the reviewer’s comment that this point was not made clear enough. In order to address this comment, we have added the following sentence in our revised manuscript (Page 12): “A single set of parameters in Derjaguin’s model ($S = 0.069$ J/m² and $\xi = 0.24$ nm), fitted against a single experimental adsorption isotherm shown in Fig. 5A, allows reproducing, without any additional fitting and yet with a very good agreement, the experimental data obtained with other pore sizes (Fig. 5B). Note that both the parameters S and ξ are obtained from a single fit as there is only one combination that allows capturing quantitatively the adsorption (low pressure) and capillary condensation (high pressure) regimes.”

Comment 8 (minor). *Code availability: version of Towhee is not mentioned. Moreover, since it is a widely used open-source code, it would be helpful for a potential reader, if the input files and initial coordinates would be provided*

Reply 8. We agree with the reviewer. In this resubmission, we have specified in the code availability statement the version of TOWHEE that was used: MCCCCS Towhee - Version 7.0.6 (July 27 2013). We have also included in the supplementary information data the .car file for each of the 4 zeolite structures.

Comment 9 (minor). *Supplementary Figure 3: it would be better to represent the pressure in the log scale, also the number notation on the scale should be consistent with the article: 0.2 instead of 0,2, etc.*

Reply 9. We agree with the reviewer. We have modified Fig. 3 to use a log scale for the pressure axis (over the entire pressure range). We have also fixed the problem with the decimal in the x axis.

Comment 10 (minor). *Typos: "data where the the", "experimental, theoretical and molecular simulations which all " a noun is missing*

Reply 10. We apologize for these typos which have been corrected in the revised version of the manuscript.

REVIEWERS' COMMENTS:

Reviewer #1 (Remarks to the Author):

I'm satisfied with the response and corrections made by the authors, and I recommend publication of the article as it is.

Reviewer #2 (Remarks to the Author):

The authors have answered my questions well and I am happy with the changes made in the manuscript. I am happy with publication.

Reviewer #3 (Remarks to the Author):

The authors fully addressed all of my comments, and therefore I have no reservations to recommend this manuscript for publication.